

# Turbidity interferes with foraging success of visual but not chemosensory predators

Jessica Lunt[1] and Delbert L. Smee

Department of Life Sciences, Texas A&M University—Corpus Christi, Corpus Christi, TX, USA
[1] Current affiliation: Smithsonian Marine Station, Fort Pierce, FL, USA

## ABSTRACT

Predation can significantly affect prey populations and communities, but predator effects can be attenuated when abiotic conditions interfere with foraging activities. In estuarine communities, turbidity can affect species richness and abundance and is changing in many areas because of coastal development. Many fish species are less efficient foragers in turbid waters, and previous research revealed that in elevated turbidity, fish are less abundant whereas crabs and shrimp are more abundant. We hypothesized that turbidity altered predatory interactions in estuaries by interfering with visually-foraging predators and prey but not with organisms relying on chemoreception. We measured the effects of turbidity on the predation rates of two model predators: a visual predator (pinfish, *Lagodon rhomboides*) and a chemosensory predator (blue crabs, *Callinectes sapidus*) in clear and turbid water (0 and ~100 nephelometric turbidity units). Feeding assays were conducted with two prey items, mud crabs (*Panopeus* spp.) that rely heavily on chemoreception to detect predators, and brown shrimp (*Farfantepenaus aztecus*) that use both chemical and visual cues for predator detection. Because turbidity reduced pinfish foraging on both mud crabs and shrimp, the changes in predation rates are likely driven by turbidity attenuating fish foraging ability and not by affecting prey vulnerability to fish consumers. Blue crab foraging was unaffected by turbidity, and blue crabs were able to successfully consume nearly all mud crab and shrimp prey. Turbidity can influence predator–prey interactions by reducing the feeding efficiency of visual predators, providing a competitive advantage to chemosensory predators, and altering top-down control in food webs.

## INTRODUCTION

Predators may affect prey populations and communities through both direct (e.g., consumption) and indirect effects (e.g., changes in prey behavior, *Trussell, Ewanchuk & Bertness, 2003*; *Preisser, Bolnick & Benard, 2005*; *Webster & Weissburg, 2009*; *Weissburg, Smee & Ferner, 2014*). These effects can cascade through communities by causing changes in behavior, density, and distributions of multiple trophic levels (*Sih et al., 1985*; *Sih, Englund & Wooster, 1998*; *Menge, 2000*; *Werner & Peacor, 2003*). The outcomes of predatory interactions are largely influenced by the ability of predators and prey to detect and respond to one another (*Powers & Kittinger, 2002*; *Weissburg, Smee & Ferner, 2014*).

Corresponding author
Jessica Lunt,
Jessica.H.Lunt@gmail.com

Perceiving a potential consumer or prey item before being detected offers a perceptive advantage that influences which organism will prevail in a given encounter (*Powers & Kittinger, 2002*; *Smee, Ferner & Weissburg, 2010*). When predators possess a perceptual advantage over prey, direct effects should be prevalent as predators should more often prevail in a given encounter. Likewise, prey can successfully avoid predators when they have a sensory advantage over predators and can detect and avoid them before being consumed. In these situations, indirect effects are likely to be prevalent.

Detection of potential predators and/or prey can be strongly affected by environmental variables that alter the sensory abilities of both predators and prey or conceal prey from predators (*Powers & Kittinger, 2002*; *Smee & Weissburg, 2006*; *Smee, Ferner & Weissburg, 2010*). Predation may increase when the environment enhances predator detection of prey and/or compromises the ability of prey to detect and avoid consumers (*Weissburg & Zimmer-faust, 1993*; *Ferner, Smee & Weissburg, 2009*; *Robinson, Smee & Trussell, 2011*). Alternatively, environmental conditions may attenuate predation by interfering with predator foraging or enhancing prey avoidance ability (*Smee, Ferner & Weissburg, 2010*). In situations where both predators and prey are affected by the same environmental conditions, and these conditions minimize the sensory abilities of both species, top-down forcing is likely to decline and the effects of predators on prey populations may shift from a combination of direct and indirect effects to exclusively direct effects as encounters become random (*Van de Meutter, de Meester & Stoks, 2005*). However, many species use multiple sensory systems which may mitigate environmental forces to some extent. Understanding how environmental variables influence sensory abilities of predators and prey will yield insights into mechanisms that influence the nature and strength of predator effects (*Weissburg, Smee & Ferner, 2014*).

In freshwater systems, turbidity as low as 20 nephelometric turbidity units (NTU), a measure of light penetration, can diminish visual acuity and decrease prey capture success and competitive interactions (*Hazelton & Grossman, 2009*). This decrease in predator efficiency may make turbidity a predation refuge from predators which are predominantly visual (*DeRobertis et al., 2003*; *Engström-Öst, Öst & Yli-Renko, 2009*). In contrast, turbidity would not likely interfere with foragers that predominantly use non-visual senses and might actually increase predation if it compromised a prey's ability to avoid predators or caused an increase in abundance of primarily chemosensory predators through meso-predator release (*Rodríguez & Lewis, 1997*; *Ritchie & Johnson, 2009*; *Lunt & Smee, 2014*).

Turbidity is increasing in coastal environments worldwide because of anthropogenic factors (*Sanden & Hakansson, 1996*; *Fujii & Uye, 2003*) mainly through increased erosion (*Khan & Ali, 2003*) and nutrient loading (*Candolin, Engström-Öst & Salesto, 2008*). Both sources affect species composition (*Khan & Ali, 2003*; *Candolin, Engström-Öst & Salesto, 2008*), though the source of turbidity can be important in determining effects on communities (*Radke & Gaupisch, 2005*). Depending on the source of turbidity the increase can be sudden (erosion during a storm) or gradual (bloom formation) and can either be long term (harmful algal blooms) or short term (sediment resuspension). Within Texas bays turbidity is primarily wind driven and can differ on small spatial scales (*Lunt &*

*Smee, 2014*). The Aransas Bay system experiences a large range of turbidity values (1-900 NTU) but averages 20 NTU, which can be considered low turbidity for marine systems (TPWD data; *Minello, Zimmerman & Martinez, 1987*; *Lunt & Smee, 2014*). Local animals therefore are subjected to variable turbidity levels within small spatial areas, depending on environmental conditions such as wind and flow that can affect their foraging efficiency.

Turbidity can influence the outcomes of predator–prey interactions in both freshwater and marine systems by altering perceptive ability (*Minello, Zimmerman & Martinez, 1987*; *DeRobertis et al., 2003*; *Sweka & Hartman, 2003*; *Webster et al., 2007*; *Ohata et al., 2011*). Moderate turbidity may enhance feeding efficiency of visual predators by providing increased contrast (*Liljendahl-Nurminen, Horppila & Lampert, 2008*), though past a certain level feeding efficiency will decrease. In addition, the effects of turbidity on the outcomes of predatory interactions may depend upon the extent to which the affected organism can use other sensory modalities to offset reductions in vision in turbid environments (*Minello, Zimmerman & Martinez, 1987*; *Abrahams & Kattenfeld, 1997*; *DeRobertis et al., 2003*; *Radke & Gaupisch, 2005*). Previously, the abundance of fish and crabs was found to be significantly affected by turbidity with fish being more abundant in low (<30 NTU) turbidity areas and crabs in high (>30 NTU) turbidity (*Lunt & Smee, 2014*). These changes in predator type altered predation efficiency: fish predation decreased with increasing turbidity whereas crab predation increased with increasing turbidity (*Lunt & Smee, 2014*). We hypothesized that turbidity influences predator–prey interactions by offering a perceptive advantage to non-visual species and alleviating predation pressure by fish on them. To test this hypothesis, the predation efficiency of a visual predator (pinfish, *Lagodon rhomboides*; *Luczkovich, 1988*) and a chemosensory predator (blue crabs, *Callinectes sapidus*; *Keller, Powell & Weissburg, 2003*) foraging on brown shrimp, (*Farfantepenaus aztecus*) or mud crabs (*Panopeus* spp.) in both low (0 NTU) and high (100 NTU) turbidity was tested in mesocosms. Shrimp use both visual and chemosensory cues to detect predators (*Minello, Zimmerman & Martinez, 1987*), while mud crabs use chemosensory means of risk detection (*Grabowski & Kimbro, 2005*; *Hill & Weissburg, 2013*). Pinfish and blue crabs were chosen because they are the most abundant fish and crab species collected by The Texas Parks and Wildlife Department and their abundances were affected by turbidity in an analysis of an 18 year data set from Texas Parks and Wildlife Department (Fig. 1; *Lunt & Smee, 2014*).

## MATERIALS AND METHODS

### Mesocosms

The study was conducted in outdoor mesocosms at Texas A&M University —Corpus Christi. The mesocosms consisted of 16 opaque, grey, polyethylene tanks with lids (61 cm × 47 cm × 41 cm). Tank lids had small windows covered with Vexar mesh to allow light into the tank while preventing species from escaping. Each tank contained 68 L of artificially created seawater at a depth of ∼0.36 m, salinity of 20 ppt, and an Aqueon™ aquarium filter and Oceanic® 250 gallon per hour aquarium pump. The filter and pump were used to aid in water circulation and to keep sediments suspended

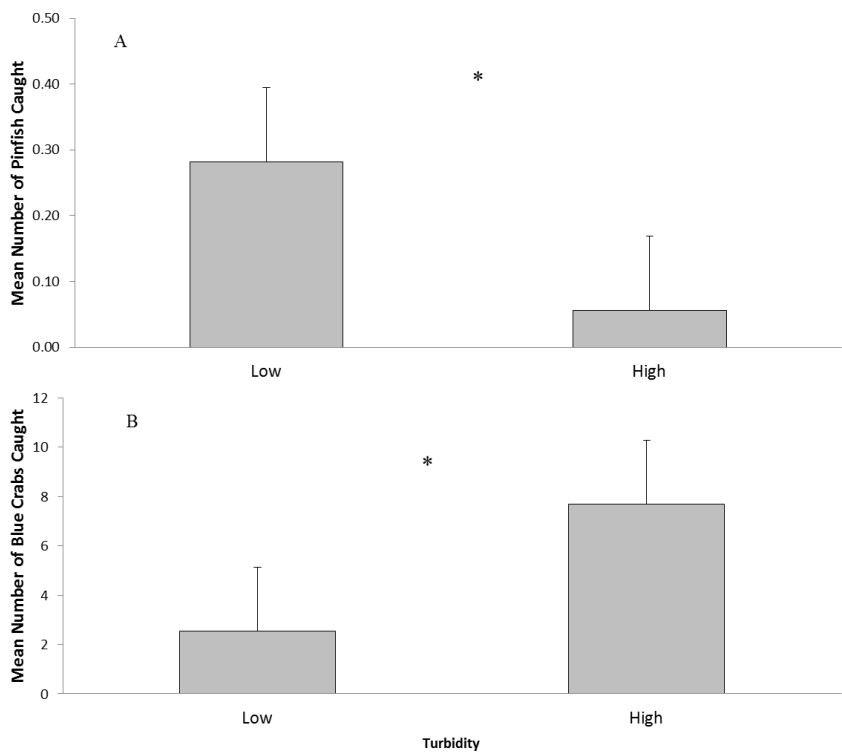

**Figure 1 Abundance of pinfish and blue crabs in Texas bays.** Texas Parks and Wildlife Department data on pinfish (*Lagodon rhomboides*) and blue crab (*Callinectes sapidus*) abundance. (A) Mean number (±SE) of pinfish caught in low (<30 NTU) and high (>30 NTU) turbidity. (B). Mean number (±SE) of blue crabs caught in low (<30 NTU) and high (>30 NTU) turbidity.

in the turbidity treatments. Flow in Aransas Bay has been measured and can range from 1–56 cm/s, thus, the currents used in the tanks are within the range of naturally occurring flow conditions (*Lunt, 2014*). Turbid treatments were created by adding 235 mL of finely ground kaolinite clay to the tanks with stirring prior to addition of animals. Kaolinite is an inert clay successfully used in previous turbidity research to mimic turbidity caused by suspended sediments (*Minello, Zimmerman & Martinez, 1987*). Preliminary trials using a Hydrolab DataSonde™ indicated that the pumps were effective at maintaining the turbidity at the treatment level for 72 h, which was the duration of our experiment. Therefore, measurements were not taken during trials to prevent the addition of the instrument from affecting the behavior of experimental animals. Turbidity was visually assessed twice daily to ensure that the pumps were working and the water appeared cloudy. Sediments were not provided in the experimental tanks as sediment can affect predation efficiency (*Minello, Zimmerman & Martinez, 1987*). Pumps were used in both clear and turbid treatments.

The model food web consisted of two predators foraging on one of two prey species. Predators used were pinfish (*L. rhomboides*; 125–188 mm total length) and blue crabs (*C. sapidus*; 100–130 mm carapace width), which forage using visual and chemosensory cues respectively. These predators used are omnivorous, estuarine species, and known for
**Table 1 Diagram of the experimental setup.**

| Predator | High (100 NTU) | | | | | | | | Low (0 NTU) | | | | | | | |
| --- | --- | --- | --- | --- | --- | --- | --- | --- | --- | --- | --- | --- | --- | --- | --- | --- |
| | Crab | | Fish | | Mix | | Control | | Crab | | Fish | | Mix | | Control | |
| Prey | MC | S | MC | S | MC | S | MC | S | MC | S | MC | S | MC | S | MC | S |
| Replication | 7 | 5 | 10 | 12 | 6 | 8 | 4 | 6 | 6 | 5 | 10 | 12 | 7 | 9 | 4 | 8 |

their voracious eating habits (*Laughlin, 1982*; *Montgomery & Targett, 1992*). Blue crabs of the size used in this study are predominantly carnivorous consuming a wide array of bivalve, gastropod, and crustacean prey (*Laughlin, 1982*). Pinfish diets vary more widely than do blue crab diet with up to 90% of pinfish diets composed of seagrass (*Hansen, 1969*; *Stoner & Livingston, 1984*; *Montgomery & Targett, 1992*). However, the proportion of seagrass in a pinfish's diet seems to be based on opportunity as seagrass is harder digest and has less energy content than meatier prey items (*Montgomery & Targett, 1992*). Pinfish consumed mud crabs and shrimp in preliminary tests prior to beginning experiments. Both predator species are abundant and were collected locally. A chemosensory (mud crabs, *Panopeus* spp.; 10–15 mm), and visual and chemosensory (brown shrimp, *F. aztecus*; 70–100 mm) prey species were used to investigate the effect of turbidity on both predators and prey. All organisms were used within 24 h of collection and in only a single trial before being returned to the site of collection (except for the prey consumed during the trials; TAMUCC IACUC 07-07).

## Feeding assays

Mesocosm experiments were set up in a 4 × 2 factorial design with 4 predator treatments and 2 turbidity levels (Table 1). Predator treatments included: no predator control, blue crab (2 crabs), pinfish (2 fish) and mix (1 fish and 1 crab). The mix treatment was performed to determine if there was any interference between predator type or if there were additive effects of predation. These treatments were performed in low (0 NTU) and high (100 NTU) turbidity levels. We elected to use 100 NTU as our turbid treatment because this value was often recorded in turbid field sites (*Lunt, 2014*) and was easier to maintain than lower levels of turbidity. Predator and turbidity treatments were interspersed. In the mesocosms, either 8 mud crabs or 4 brown shrimp were added as prey, but not both simultaneously. Predators were allowed to forage on prey for 72 h. At the end of each trial, the number of prey eaten was recorded. No blue crabs or pin fish perished during the study.

## Analysis

Differences in the number of eaten prey between predator and turbidity treatments were analyzed using a 2-way ANOVA with predator and turbidity treatments as fixed factors (*Sokal & Rohlf, 1995*). Assumptions for ANOVA were tested using diagnostic plots (*Sokal & Rohlf, 1995*). Pairwise differences of all possible predator and turbidity combinations were compared using a simple main effects test (*Kirk, 1982*). Uneven sample sizes (Table 1) resulted because of animal availability.
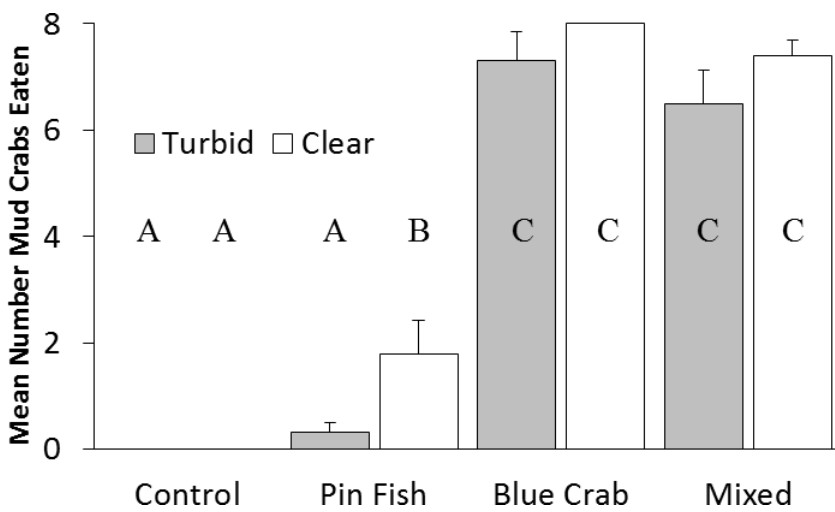

**Figure 2 Mud crabs eaten.** Mean number (±SE) of mud crabs eaten in turbid and clear 391 treatments. Turbidity ($p < 0.05$) and predator treatment ($p < 0.001$) were significant factors in a 392 two-way ANOVA. The interaction term was not significant ($p = 0.54$). Letters denote significant 393 pairwise differences.

## RESULTS

Predation on mud crabs was affected by both the predator type ($F_{7,45} = 130.4, p < 0.0001$) and by turbidity ($F_{7,45} = 4.94, p = 0.03$). The interaction between turbidity and predator type was not significant ($F_{7,45}\ 0.73, p = 0.54$). When blue crabs were present, all mud crabs were eaten in clear water and nearly all in the turbid treatment. Pairwise differences between treatments revealed that turbidity only had a significant effect on pinfish foraging (Fig. 2). Similarly, the number of shrimp consumed was affected by predator type ($F_{7,57} = 164.4, p < 0.001$) and turbidity ($F_{7,57} = 7.32, p < 0.001$). The interaction term was not significant ($F_{7,57} = 1.91, p = 0.14$). Blue crabs consumed all shrimp in clear water and nearly all in turbid water. Pairwise differences between treatments revealed that turbidity only had a significant effect on pinfish foraging (Fig. 3).

## DISCUSSION

Visual acuity in freshwater and marine fishes can be compromised by turbidity, reducing their foraging efficiency (*Minello, Zimmerman & Martinez, 1987*; *Macia, Abrantes & Paula, 2003*; *Aksnes et al., 2004*; *Aksnes, 2007*). Turbidity can influence both predation rates and the type of predator effect (direct vs. indirect) (*Abrahams & Kattenfeld, 1997*; *Van de Meutter, de Meester & Stoks, 2005*). For example, Atlantic Cod (*Gadus morhua*) reacted more slowly to predatory threats and took longer to forage on mysid shrimp as turbidity increased (*Meager et al., 2005*). Yet, turbidity may interact with other factors such as substrate complexity, sediment type, and prey density to influence the outcome of predator–prey interactions (*Minello, Zimmerman & Martinez, 1987*; *Macia, Abrantes & Paula, 2003*). For example, thorn fish (*Terapon jarbua*) predation on white shrimp (*Penaeus indicus*) declined as turbidity increased, but, thorn fish predation on brown shrimp (*Metapenaeus monoceros*) was influenced by sediment and prey density in

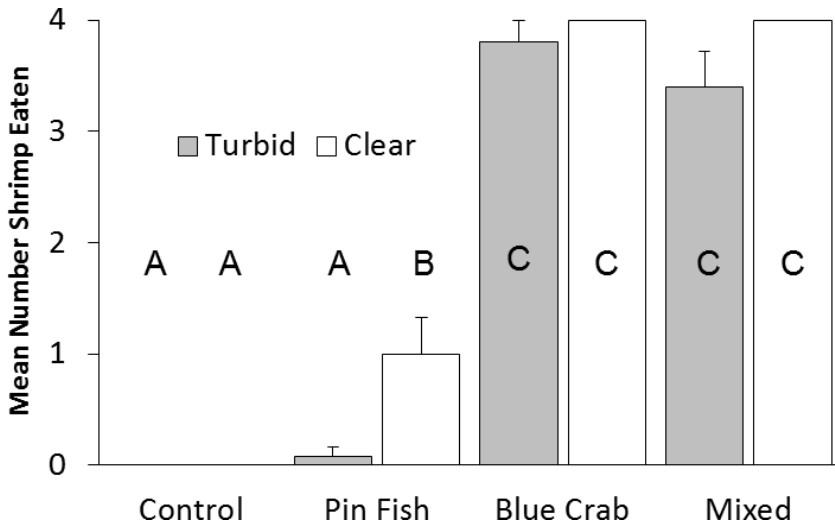

**Figure 3 Brown shrimp eaten.** Mean number (±SE) of brown shrimp eaten in turbid and clear treatments. Turbidity ($p < 0.01$) and predator treatment ($p < 0.001$) were significant factors in a two-way ANOVA. The interaction term was not significant ($p = 0.14$). Letters denote significant pairwise differences.

addition to turbidity so that predation was highest at intermediate turbidity levels (*Macia, Abrantes & Paula, 2003*). The effects of turbidity on foraging by three predatory fish: southern flounder (*Paralichthys lethostigma*), pinfish, and Atlantic croaker (*Micropogonias undulatus*) preying upon brown shrimp provided with different substrates produced variable results (*Minello, Zimmerman & Martinez, 1987*). Their findings indicated that turbidity decreased flounder predation, increased croaker predation, and both increased and decreased pinfish predation depending upon substrate type. To focus solely on the effects of turbidity on pinfish and blue crabs, we elected not to use substrate in our experiments. Consistent with the earlier studies described above, we found that turbidity inhibited predation by a visual predator (pinfish) on both mud crabs and brown shrimp.

Turbidity, particularly at the levels used in this study, interferes with light penetration and would hinder vision, but it is unlikely to inhibit other sensory modalities (*Eiane et al., 1999*; *Ohata et al., 2011*). Thus, organisms that forage by tactile cues or chemoreception may be unaffected by turbidity, and may gain a competitive advantage in turbid waters over competitors that forage using visual cues (*Eiane et al., 1999*). This hypothesis is supported by the results of our study as blue crabs were unaffected by turbidity, and consumed nearly all mud crabs and shrimp in all treatments in which they were present. In Norwegian fjords, jellyfish abundance is highest when light penetration is lowest. This is attributed to fishes being unable to effectively forage and acquire enough energy to maintain their populations while jellyfish, as tactile foragers, were unaffected by turbidity (*Eiane et al., 1999*). The interaction between turbidity and chemosensory foragers may be more complex in natural systems. Suspended particles may adsorb chemical components of natural exudates and therefore decrease chemosensory abilities in natural systems.

When turbidity alters the abundance or effectiveness of predators, cascading effects in aquatic food webs occur. The abundance of fish and their foraging rates decline in turbid environments (*Eiane et al., 1999*; *Aksnes et al., 2004*; *Lunt & Smee, 2014*). *Eiane et al. (1999)* and *Aksnes et al. (2004)* both noted changes in zooplankton communities in turbid environments and attributed this to alterations in predation by fish. In the Gulf of Mexico, turbidity was found to switch food webs from being dominated by fish to being dominated by crabs (*Lunt & Smee, 2014*). In this area, fish predation on crabs was reduced when turbidity exceeded 30 NTU in the field, and both mud crabs and shrimp were more abundant on oyster reefs when turbidity was above 30 NTU (*Lunt & Smee, 2014*).

We tested the hypothesis that turbidity reduces fish ability to forage, thereby releasing lower trophic levels (such as crabs) from top-down control (*Lunt & Smee, 2014*). Pinfish were less successful consumers in high turbidity and consumed significantly fewer crab and shrimp prey in these conditions. These results mirror previous studies using freshwater organisms in which predation by visual predators declined in elevated turbidity (*DeRobertis et al., 2003*; *Sørnes & Aksnes, 2004*; *Engström-Öst, Öst & Yli-Renko, 2009*). Reduced consumption in turbid treatments by pinfish is likely a result of their reliance on vision to forage. Mud crabs likely have a sensory advantage in turbid conditions, escaping detection by pinfish by being able to detect fish chemical cues to avoid them. Brown shrimp are more active in turbid treatments, but, were not more vulnerable to pinfish predation in turbid conditions in our study, perhaps because they can also use chemical cues to detect and avoid pinfish.

Blue crabs are known to be voracious predators, and effectively consumed all prey items in both clear and turbid treatments. Even in mixed assemblages with one blue crab and one pin fish, predation rates were consistently above 80%, even in turbid treatments when fish foraging was compromised. Crabs forage primarily through chemoreception, which would not be affected by increased turbidity at the levels used in this study (*Eiane et al., 1999*; *Ohata et al., 2011*). Blue crabs are also a prey species to many fish and bird species and may seek out turbidity as a refuge from these consumers (*DeRobertis et al., 2003*; *Engström-Öst, Öst & Yli-Renko, 2009*), thereby increasing their abundance in high turbidity sites (*Lunt & Smee, 2014*). The effects of turbidity on foraging efficiency of visual predators but not chemosensory predators helps explain the reduction in fish and increase in crab abundance when turbidity increases (*Lunt & Smee, 2014*).

## ACKNOWLEDGEMENT

We thank the members of the TAMUCC Marine Ecology Lab and the Marine Ecology Class of Fall 2012 for logistical support.

### Funding

This research was supported by Texas Sea Grant, Texas Research Development Fund, a Texas A & M University-Corpus Christi Faculty Enhancement Grant, the Harvey Weil Sportsman Conservationist Award, and the Ruth A. Campbell Professorship to DL Smee. The funders had no role in study design, data collection and analysis, decision to publish, or preparation of the manuscript.

### Grant Disclosures

The following grant information was disclosed by the authors:
Texas Sea Grant.
Texas Research Development Fund.
Texas A & M University-Corpus Christi Faculty Enhancement Grant.
Harvey Weil Sportsman Conservationist Award.
Ruth A. Campbell Professorship.

### Competing Interests

The authors declare there are no competing interests.

### Author Contributions

- Jessica Lunt conceived and designed the experiments, performed the experiments, analyzed the data, wrote the paper, prepared figures and/or tables, reviewed drafts of the paper.
- Delbert L. Smee conceived and designed the experiments, analyzed the data, contributed reagents/materials/analysis tools, wrote the paper, prepared figures and/or tables, reviewed drafts of the paper.

### Animal Ethics

The following information was supplied relating to ethical approvals (i.e., approving body and any reference numbers):

TAMUCC Institutional Animal Care and Use Committee 07-07.

### Supplemental Information

Supplemental information for this article can be found online at http://dx.doi.org/10.7717/peerj.1212#supplemental-information.

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
