# Peer review of "Turbidity interferes with foraging success of visual but not chemosensory predators"

_PeerJ, doi:10.7717/peerj.1212_

## Round 0.1 · original submission · Major Revisions

Dear Authors,

two colleagues have reviewed your manuscript. Both of them appreciated your work, but also expressed some concerns that need your attention. In particular, more experimental details and ecological data are necessary.

Reviewer 1 ·

Basic reporting

As far as I see, the ms complies with requirements

Experimental design

Authors should justify the selection of the species they used on ecological arguments.

They could provide more information about the relevance of experimental treatments with natural conditions encountered by these organisms

The relevance of one experimental treatment (mix) is not obvious and should be explained

Validity of the findings

Sample sizes should be specified to better assess the power of the tests

While sudden changes in turbidity have been show to affect interspecific interactions (cichlid of the Lake Victoria) authors seem to mainly focus on estuarine species where long term evolutionary process may have shaped the communities. This is confusing and should be avoided or replaced in an appropriate time framework

Additional comments

The authors investigated the effect of turbidity on foraging efficiency of predators using two sensory modalities vision and olfaction. They found that turbidity generally reduced foraging performance but that the visual predator was much more affected when vision was impaired than the olfactory predator.
The general idea of testing the relative performance of predators with different perceptual abilities is interesting. It is not very clear whether this has been done before on other species and some hints about its novelty/complementarity should be given for the reader even if it is not an evaluation criterion as such for the journal.
More generally, I think the introduction may benefit from refocusing. The framework is about how environmental conditions shape trophic networks and in particular select functional traits, sensory abilities for the current system. This seems to me a more integrative approach that would allow the comparison with other ecosystems than estuaries (think of deep sea or underground environments). Then, it would be possible to develop the focus on the specific topic of the manuscrit. It may also be useful to link paper with the general theories of sensory ecology.
The experiment set is simple and the results are clear. I have one concern. There is no information about the ecology of the species’ used here. Are these specialist of estuarine environments or even exposed to such high level of turbidity? Why were these species selected in the first place? This information is necessary to analyse the results and assess the validity of the discussion. I have additional comments about methological choices. In particular, it is not really clear why testing the mix treatment (crab+fish) would bring insight to the question. Also, authors should justify more clearly the selection of turbidity level so that the reader can linked them to actual conditions encountered by organisms (see below). Finally, it is should be make clear whether water agitation, necessary to maintain turbidity, is likely to affect the behaviour of predators and preys in a way that would not reflect natural conditions.

Specific comments
L43 probably “perceptual” instead of “sensory”
I would rather use “indirect” effect of predation instead of “non lethal” effect
L49 or conceal from predators
L54-57 May be too simplistic. Environmental conditions can limit detection on a particular sensory channel but not on every channel available to an organism or to an alternate predator. Therefore, the assumption about the lower top-down forcing may not be met. I think you may focus more on the selection of functional traits across the trophic network in particular in species involved in prey-predation relationships.
L61-68 You may also refer to species that use electric sensory organs for prey detection and foraging. There are good examples of electric fishes in very turbid and very large rivers like the Amazon.
L68-73 This is the primary issue. If prey or predator cannot cope with turbidity a particular structure of the trophic network (including functional traits) can arise.
L84 Why such extreme values? It may be good to give a bit more information about the turbididity regime for instance the annual median and quantiles, if these values exist for the area where experimental preys and predators come from. Why did not you choose to test one or two intermediate treatments?
L93 Please give the dimensions of the tank. What was the colour of the tanks as background could affect the contrast with prey colour, and thus detection, it is worth giving this information if you use a visual predator?
L92-101 I understand why water circulation was used to keep kaolinite suspended. Could you precise whether this level of water movement is naturally encountered both by preys and predators in natural conditions? If not, could these novel conditions affect predation rate (fish may may more disturbed in anormally agitated waters), which would limit the extrapolation of conclusions?
L92-101 Did you check turbidity level througout the experiment. From my own experience, clay deposit on the bottom of the tanks even when using water pumps, at least if you do not use “tsunami grade” pumps? Given, the initial difference between your treatments this should not be an issue though.
L103-110 Could you give more information about how frequently these preys are consumed by both predators and in which natural conditions encounters occur (turbidity range)? Are they both common preys in each predator’s diet?
L111-120 How many trials have been performed? How many crabs and how many fishes were tested? It is hard to tell what your sample size and thus the power of your test. According to your results, turbidity seems to affect both predators but not equally, the crab being more efficient regardless of the treatment.
L114 Why did you test the “mix” treatment? What was your prediction? I do not see clearly the rationale.
L114 why did you these experimental levels (0 and 100 NTU). There is no information about the turbidity regime encountered by these populations so that it is hard to tell the biological meaning ot either treatment. Could you provide either statistics or a graph of the annual variation of turbidity? It may have been more informative to compare a control (0 NTU) with a high value of turbidity (90th percentile for instance) and a regular condition (median). One can question the actual effect on prey-predator relationships if the high turbidity level occurs rarely.
L125 Could you precise whether your data and residuals meet the assumptions of ANOVA?
L156-157 Please refer to these earlier studies
L158-160 Although I agree on general terms, I would be less affirmative. It is likely that some suspended particles causing turbidity interact with organic compounds released by organisms. Clays can adsorb peptides for instance. Thus, I would expect detection to be lowered in natural conditions even for predators using chemodetection. Remember you used a small tank in a simple chemical environment (kaolinite only) in which the degradation of chemical cues would be reduced.
L161 Here again, moderate your argument.
L179 What novel information bring your own study? You did not make that point clear
L182 “chemical cues” may be more appropriate than “exudates”
L191-192 Why would this increase their abundance in high turbidity sites? It is possible that turbid sites are the main habitat for this species. There is not enough information about the species’ecology to tell.
L195-196 and L205 “proliferate” and “decimate” could be replaced by more neutral terms
L203-213 I think the concluding paragraph is awkward. One cannot compare bluntly short-term and long term processes. I suspect the cited example of shark overfishing to concern a very recent period. It is not a surprise that sudden and strong events on top predators have cascading effects on the trophic network. In your work you address, foraging performance in estuarine environments where communities have formed over a long period as a response to this particular environment. One can expect selection of species with particular functional traits. Olfaction should prevail over vision for any function foraging or mating.

·

Basic reporting

Fine, refer to reviewer's comments attachment

Experimental design

Fine

Validity of the findings

Fine

Additional comments

Review of ” Turbidity interferes with foraging success of visual but not
chemosensory predators”

I am Damian Moran and I choose to give an open review.

General comments:
The manuscript by Lunt & Smee investigates the effect of turbidity on fish and crab predation rates. There are two types of prey. The predators and prey have different sensory strengths and the tank based experiments are designed to measure predation efficiency in relation to turbid and non turbid environments. The experiments are rather simply done with what I suspect are fairly low sample sizes, but these experiments are all that are probably needed to establish what are fairly stark differences in predation rates. The manuscript focuses a lot on sensory biology without having done sensory biology experiments (e.g. establishing dose-response in terms of stimuli intensity, properly separating effects of light intensity versus scattering, measuring light transmission, strike distances, separating the effect of the turbidity treatment from effects on vision, having blue crabs predate upstream of prey to check chemosensory capabilities etc). The discussion focuses mainly on ecosystem level consequences of the results rather than the sensory biology of the study organisms, and this seems to be due to the interest the authors have on the former topic. Besides a few minor comments I don’t have any problems with the data or analysis. The expansive discussion is not my style, but the core of this small study is pretty solid.

Specific comments:

1) In my opinion, the authors have not characterised the nutritional ecology of the of the pinfish very well at all. According to

Montgomery & Targett J. Exp. Mar; Bioi. Ecol., 158 (1992) 37-57

”The pinfish Lagodon rhomboides (L.) is an abundant omnivore in temperate zone and subtropical seagrass meadows along the Atlantic and Gulf coasts of North America(see Darcy, 1985; Huh, 1986). It is a major consumer of vegetation in these meadows (Darcy, 1985). Studies have reported that seagrasses (i.e., Thalassia testudinum, Syringodium filiforme, and Zostera manna L.) constitute 18-90% of the diet of the pinfish (Hansen, 1969; Adams, 1976; Stoner; 1980; Stoner & Livingston, 1984). Macroalgae (primarily green algae) are also important, constituting as much: as 50% of the pinfish diet (Darnell, 1958; Hansen, 1969; Stoner, 1980) mentioned”

It would appear these fish are more at least as much herbivore as carnivore, and this might play a significant role in their predatory drive or ability to access nutrition in an area of increased turbidity. If the authors are extending their findings to the ecosystem level (which they do in the Discussion, extensively), then there is an obligation to consider the total nutritional ecology of the species.

2) Changes in turbidity may be advantageous to certain life stages but not at others. For example, increased turbidity can be an advantageous to planktonic predators by increasing the contrast prey.

Boehlert, G. W., and J. B. Morgan. 1985. Turbidity enhances feeding abilities of larval Pacific herring Clupea harengus pallasi. Hydrobiologia 123:161–170.

When culturing marine fish larvae clay or microalgae are usually added to the culture tank to improve predation rates. Perhaps it is worth mentioning so that readers can get a sense of the complex relationship between vision, light transmission and turbidity. Turbid may not always equal bad for visual predation, context is important.

3) Line 85: Change to …chemosensory means of risk detection

4) Materials and Methods
The experimental structure needs to be better explained. The way I read it at the moment is:
There were 2 types of prey, 2 types of predators, and 2 levels of turbidity. The predators were tested in 4 combinations: no predators, pred 1 only, pred 2 only, pred 1+2. The prey were also tested separately. So the design seems to be 4 predator treatments x 2 turbidity treatments x 2 prey treatments = 16 treatment matrix.

How many replicate trials were performed per treatment? What was the rationale for choosing that sample size? It looks like there were a different number of trials with mud crab versus shrimp.

5) Results
L130-131: ”Pairwise differences between treatments revealed that turbidity had a significant effect on pinfish foraging, but not in the other treatments (Figure 1).”

The clause ”but not in the other treatments” is clumsy due to the on-in preposition change from the previous clause. Perhaps simplify (and the other sentence that reads the same later) to:

Pairwise differences between treatments revealed that turbidity only had a significant effect on pinfish foraging.

6) L192-194: ”The effects of turbidity on foraging efficiency of visual predators but not chemosensory predators helps explain the reduction in fish and increase in crab abundance when turbidity increases (Lunt & Smee, 2014).”

From my experience in aquaculture, I have learned that there are many skin and gill related pathologies associated with a high suspended solids load in the water. Could an auxiliary reason that fish avoid turbid areas be due to gill or skin irritation? High levels of dissolved organic material are also associated with lower dissolved oxygen content. Would non-feeding related factors also play a role in the observed reduction in fish density? Or do the authors feel the evidence points to distributional patterns all being down to visual predation capacity?

The Discussion does venture out into ecosystem wide effects from what is a fairly small controlled tank study (that’s fine with me), but the corollary to such a venture is that a wide range of controlling factors should be considered.

7) Figure 1 caption: (+SE) written twice

---

## Round 0.2 · accepted · Accept

Dear authors, thank you very much for your effort in improving your manuscript.

·

Basic reporting

My reviewer comments have been adequately addressed

Experimental design

My reviewer comments have been adequately addressed

Validity of the findings

My reviewer comments have been adequately addressed

Additional comments

My reviewer comments have been adequately addressed